# Analysis of Complete Chloroplast Genome: Structure, Phylogenetic Relationships of *Galega orientalis* and Evolutionary Inference of Galegeae

**DOI:** 10.3390/genes14010176

**Published:** 2023-01-09

**Authors:** Junjie Feng, Yi Xiong, Xiaoli Su, Tianqi Liu, Yanli Xiong, Junming Zhao, Xiong Lei, Lijun Yan, Wenlong Gou, Xiao Ma

**Affiliations:** 1College of Grassland Science and Technology, Sichuan Agricultural University, Chengdu 611130, China; 2Sichuan Academy of Grassland Science, Chengdu 611130, China

**Keywords:** *Galega orientalis*, Galegeae, chloroplast genome, phylogenetic relationship

## Abstract

*Galega orientalis*, a leguminous herb in the Fabaceae family, is an ecologically and economically important species widely cultivated for its strong stress resistance and high protein content. However, genomic information of *Galega orientalis* has not been reported, which limiting its evolutionary analysis. The small genome size makes chloroplast relatively easy to obtain genomic sequence for phylogenetic studies and molecular marker development. Here, the chloroplast genome of *Galega orientalis* was sequenced and annotated. The results showed that the chloroplast genome of *G. orientalis* is 125,280 bp in length with GC content of 34.11%. A total of 107 genes were identified, including 74 protein-coding genes, 29 tRNAs and four rRNAs. One inverted repeat (IR) region was lost in the chloroplast genome of *G. orientalis*. In addition, five genes (*rpl22*, *ycf2*, *rps16*, *trnE-UUC* and *pbf1*) were lost compared with the chloroplast genome of its related species *G. officinalis*. A total of 84 long repeats and 68 simple sequence repeats were detected, which could be used as potential markers in the genetic studies of *G. orientalis* and related species. We found that the Ka/Ks values of three genes *petL*, *rpl20*, and *ycf4* were higher than one in the pairwise comparation of *G. officinalis* and other three Galegeae species (*Calophaca sinica*, *Caragana jubata*, *Caragana korshinskii*), which indicated those three genes were under positive selection. A comparative genomic analysis of 15 Galegeae species showed that most conserved non-coding sequence regions and two genic regions (*ycf1* and *clpP*) were highly divergent, which could be used as DNA barcodes for rapid and accurate species identification. Phylogenetic trees constructed based on the *ycf1* and *clpP* genes confirmed the evolutionary relationships among Galegeae species. In addition, among the 15 Galegeae species analyzed, *Galega orientalis* had a unique 30-bp intron in the *ycf1* gene and *Tibetia liangshanensis* lacked two introns in the *clpP* gene, which is contrary to existing conclusion that only *Glycyrrhiza* species in the IR lacking clade (IRLC) lack two introns. In conclusion, for the first time, the complete chloroplast genome of *G. orientalis* was determined and annotated, which could provide insights into the unsolved evolutionary relationships within the genus Galegeae.

## 1. Introduction

As the main semi-autonomous genetic organelle in plant cells, chloroplasts contain their own independent DNA (cpDNA) and play an important role in carbon fixation, oxygen production and regulation of many metabolic pathways [1]. Chloroplast (cp) genome is typically a closed loop loaded structure, usually ranging in size from 115 to 180 kb, which consists of a large single-copy region, a small single-copy region and a pair of inverted repeats (IRs) regions in most angiosperms [2]. In comparison with the nuclear genome, cp genome has a small size, haploid nature, maternal inheritance, low rate of nucleotide substitution and highly conserved genome structure [3]. There are striking variations in genome size, gene number, and GC content in different plant lineages’ cp genomes, as well as signatures of selection, genomic rearrangement, intron gains and losses, and expansion/contraction of the IR [4]. Comparative analyses of the chloroplast genomes played a crucial role in understanding the evolution of the chloroplast genome [5,6,7,8].

Fabaceae, the third largest family of angiosperms after Orchidaceae and Asteraceae, contains many species with wide distribution [9,10]. Currently, Fabaceae is classified into six subfamilies, namely Caesarpinioideae, Certifiidoideae, Detarioideae, Dialioideae, Duparquettioideae and Papilionoideae, according to a new classification system based on cp *matK* gene [11]. Among them, Papilionoideae is an excellent model for evolutionary analysis and investigation of cp genomic structure variation because of numerous rearrangements, gene deletions, intron deletions, and local hypervariable regions present within this taxon [12]. As a result, Papilionoideae has been divided into seven main evolutionary clades, including Cladrastis, Genistoids, Dalbergioids, Mirbelioids, Millettioids, Robinioids and the IR lacking clade (IRLC). Notably, IRLC is considered a useful unit for studying the evolutionary mechanism of Papilionoideae due to its high nucleotide mutation rate and obvious structural variation [13]. In the IRLC lineage, cp genomes were found rearranged in many species, including gene replication, loss, and sequence inversion, which is particularly common in *Trifolium* [14]. However, the cause of these rearrangements in the cp genome is unclear.

*Galega orientalis* is a member of the Galegeae tribe in Papilionoideae, which belongs to the genus *Galega* along with *Galega officinalis*. It is a perennial cool-season forage legume species mainly naturally distributed in grassland and subalpine meadows at an altitude of 330–1800 m around the Black Sea region extending from southeastern Europe through the Caucasus to western Asia [15,16]. It has higher crude protein content than common forage (*Medicago sativa*, *Trifolium pratense*) and excellent disease and pest resistance [17,18], good livestock palatability and low *galegine* content [19]. In addition, *G. orientalis* is a nectar plant useful for producing excellent quality of honey, and is also used to extract *dimethylbiguanide* to treat diabetes as a medicinal plant [20,21]. Therefore, because of its high ecological value and economic value, *G. orientalis* has been introduced and promoted in many countries [22].

However, the improvement of biomass trait by transgenic technique in *Galega* species has been reported [22]. Currently, only few reports are available on the chloroplast genome sequence of the genus *Galega*, which seriously hampers our understanding of molecular identification, evolution and phylogeny of Galegeae tribe [23,24]. Especially, as the model genus of Galegeae, *G. officinalis* is the only *Galega* species with an available complete chloroplast genome [25]. Therefore, sequencing other *Galega* species would greatly enrich the cp genome of this genus. Here, we sequenced the complete chloroplast genome of *G. orientalis*, and compared it with the chloroplast genome of *G. officinalis* as well as with the cp genomes of other six genera species in Galegeae: *Astragalus*, *Calophaca*, *Caragana*, *Oxytropis*, *Tibetia* and *Glycyrrhiza*. We also elucidated the evolutionary relationships within the genus *Galega* in Galegeae based on their complete cp genomes. Our results will provide important information and markers for phylogenetic and evolutionary studies in Galegeae.

## 2. Methods

### 2.1. Genomic DNA Extraction and Sequencing

The young leaves of *Galega orientalis* were collected from Hongyuan County, Sichuan, China. DNA was extracted from leaves immediately after drying with silica gel. Total genomic DNA was extracted from 500 mg of dried leaves using the TruSeq DNA Sample Preparation Kit (Vanzyme, China). In accordance with the Illumina manual (San Diego, CA, USA), sheared low-molecular-weight DNA fragments were used to construct paired-end libraries. Sequencing of the completed libraries was performed on an Illumina HiSeq 2500 using a PE150 sequencing strategy and 300 bp insert size. The data that support the findings of this study have been deposited into CNGB Sequence Archive (CNSA) [26] of China National GeneBank DataBase (CNGBdb) [27] with accession number CNP0003740.

### 2.2. Cp Genome Assembly and Annotation

The sequence containing more than 3 unidentified bases in the raw data of *Galega orientalis* was eliminated. Sequences with Q20 less than 60% were eliminated. Low quality sequences at 3′ end, and sequences with length less than 60 bp were eliminated. Based on the reference genome of *Galega officinalis* (NC_051885.1), cp DNA was assembled. Briefly, sequences were assembled into Contigs using MetaSPAdes software [28], and non-target sequences were removed. The sequences of *G. orientalis* and *G. officinalis* were verified using NCBI BLAST, with the following parameters: E value less than 1 × 10^−5^ identity values of nearly 100%, and almost no gap. Then, CPGAVAS2 was used to annotate chloroplast genes [29]. The coding sequence (CDS) and rRNA of each organism were analyzed using BLAST Version 2.2.25, HMMER Version 3.1b2 and the NCBI genome database CP. ARAGORN V 1.2.38 [30] was used with a search server TRNAscan-SE [31] for predicting and confirming tRNA sequences. Finally, the consensus annotation was obtained using CPGAVAS2 and visualized with OGDRAW [32].

### 2.3. Characterization of Repeat Sequences

The REPuter program [33] was used to identify forward repeats, reverse repeats, complementary sequences and palindromic sequences of at least 30 bases, with hamming distances of three and a 90% identity rate. Simple sequence repeats (SSRs) were detected using the microsatellite identification tool MISA [34] with the following parameters: the minimum SSR motifs number of mononucleotide, dinucleotide, trinucleotide repeats, other nucleotides were set to 10, 5, 4, and 3, respectively.

### 2.4. Ka/Ks Calculation, Divergent Hotspots Identification and Genome Structure Comparison

In the present study, the shared protein coding genes of 14 closely related species of *G. orientalis* in Galegeae (*G. officinalis*, *Astragalus mongholicus*, *A. galactites*, *A. laxmannii*, *Calophaca sinica*, *Caragana jubata*, *C. korshinskii*, *Oxytropis diversifolia*, *O. splendens*, *O. bicolor*, *Tibetia liangshanensis*, *Glycyrrhiza zaissanica*, *G. gobica* and *G. alaschanica*) were analyzed based on Ka/Ks value using KaKs Calculator v2.0 [35]. Furthermore, to evaluate the nucleotide polymorphism (Pi) of *G. orientalis* and *G. officinalis* cp genomes, the complete chloroplast genome sequence was aligned with MAFFT v7.402 [36] and then DnaSP v.6.12 [37] software was used to perform sliding window analysis to calculate Pi values with the following parameters: window length of 800 bp and step size of 200 bp. mVISTA [38] software was used to compare the complete cp genomes of *G. orientalis* and 14 Galegeae species in the shuffle-LAGAN mode and with *G. orientalis* annotation as a reference.

### 2.5. Phylogenetic Analysis

Complete chloroplast genome sequences of 15 Galegeae species and another 16 legume species, including *Crotalaria albida* (NC_061361.1), *Crotalaria pallida* (NC_053562.1), *Piptanthus concolor* (NC_051876.1), *Piptanthus nepalensis* (NC_056139.1), *Ammopiptanthus mongolicus* (MK704436.1), *Ammopiptanthus nanus* (NC_034743.1), *Lotus corniculatus* (MT528596), *Carmichaelia australis* (MF597719), *Vicia sepium* (NC_039595), *Lupinus micranthus* (KU726828), *Clitoria ternatea* (MN709849.1), *Cicer arietinum* (NC_011163.1), *Medicago sativa* (KU321683), *Medicago hybrida* (NC_027153.1), *Onobrychis viciifolia* (MW007721.1) and *Trifolium alexandrinum* (MN857160) were used to construct a phylogenetic tree. Multiple sequence alignment of the 31 complete cp genome sequences was performed using MAFFT [34] with default parameters. Phylogenetic trees were constructed using the IQ- TREE (Maximum Likelihood Estimation) software applying *Pterocarpus santalinus* (MN251250.1) and *Arachis hypogaea* (NC_037358.1) as outgroups, with a bootstrap number of 1000 [39]. The ModelFinder module confirmed that the TVM+F+R7 was the best nucleotide substitution model.

## 3. Results

### 3.1. Chloroplast Genome Characteristics and Structure of Galega orientalis

In total, there were at least 16 million ReadSum (pair-end reads) in the original sequencing data of *Galega orientalis*, among which Q20 > 96% and Q30 > 90%. After removing adaptor sequences and low-quality sequences, 1,283,368 clean reads were remained. Both *G. orientalis* and *G. officinalis* cp genomes were found lacked IR region. The cp genome size of *G. orientalis* and *G. officinalis* was 125,280 and 125,086 bp, with an overall GC content of 34.11% and 34.18%, respectively (Figure 1 and Figure 2). Based on chloroplast genome loop map, the *G. orientalis* chloroplast showed significant similarity with that of *G. officinalis* in gene content, order, and direction.

In total, 107 genes were annotated in *G. orientalis*, including 74 protein-coding genes, 29 tRNA and four rRNA. Overall, 14 genes contained introns, of which nine encoded proteins, and five encoded tRNA. Of these, 13 genes had one intron, whereas *ycf3* had two. *ycf1* and *trnK-UUU* had the smallest intron (30 bp) and largest intron (2475 bp), respectively (Table 1). The main structural differences were as follows: there was an intron in the *ycf1* gene of *G. orientalis*, but not in *G. officinalis* (Appendix A). Furthermore, five genes, including *rpl22*, *ycf2*, *rps16*, *trnE-UUC* and *pbf1* were lost in *G. orientalis* compared with the chloroplast genome of *G. officinalis*.

Among the 107 chloroplast genes in *G. orientalis*, 11 encoded ribosomal proteins, 12 were associated with transcription, 33 were involved in self-replication, and 44 genes were associated with photosynthesis. These genes encoded photosystem I/II subunits, Rubisco subunits, ATP synthetase subunits, cytochrome b/f complexes, C-type cytochrome synthesis and NADH dehydrogenase subunits. In addition, four genes encoded maturase, protease, chloroplast envelope membrane protein and Subunit of acetyl-CoA, and normal nucleotide sequences of the three structural genes (Table 2).

### 3.2. SSRs and Repeat Sequences Analysis

Analysis of repeat sequences is important for gene rearrangement and phylogenetic construction. Therefore, we analyzed and compared the dispersed repeats in *G. orientalis* with those of *G. officinalis*. In total, 84 long repeats were found in the complete chloroplast genome of *G. orientalis*, including 52 forward repeats (30–170 bp), 26 palindromic repeats (30–72 bp), three reverse repeats (30–31 bp) and three complement repeats (30 bp). Among them, the maximum number of repeats (N = 170) was located in the ycf1-trnN-GUU region. In addition, 73.8% of the repeat sequence occurred in the intergenic spacer region (IGS), 14.3% in coding regions (*psaA*, *psaB*, *ndhF*, *ycf1*, *ycf4*, *psbA*, *rbcL* and *trnS—GCU*), 11.9% in the introns of *ycf1*, *ndhA*, *ndhB*, and *rpoC1* (Appendix A). In *G. officinalis*, 59 long repeats were found, of which 64.4% were located in IGS, 27.1% in coding regions, and 8.5% in the introns (Appendix A). *G. officinalis* had a low number of repeats, but the distribution of repeats was similar to that of *G. orientalis* (Figure 3A).

SSRs are highly polymorphic and are valuable tools for genetic diversity studies and analysis of phylogenetic relationships among plant populations. A total of 68 and 82 SSRs (ranging in size from 10 to 24 bp and 10 to 36 bp) were detected in *G. orientalis* and *G. officinalis,* respectively. In *G. orientalis*, the 68 SSRs comprised 33 mono-repeats, 20 di-repeats, two tri-repeats, 12 tetra-repeats and one penta-repeat. The 82 SSRs in *G. officinalis* included 48 mono-repeats, 21 di-repeats, two tri-repeats, seven tetra-repeats, two penta-repeats and two hexa-repeats (Figure 3B). Overall, the frequency of mononucleotide repeats was higher in *G. officinalis*. In addition, mononucleotide repeats in *G. orientalis* were predominantly of the A/T motif (96.9%). All dinucleotide repeats were composed of the AT/TA motif. In total, 43 and 21 SSRs were distributed in IGS and CDS regions of *G. orientalis*, respectively (Figure 4A). However, no C/G motif was found in *G. officinalis* (Figure 4B), but CGTCCA and CTTATA hexanucleotide were found in *ycf1* coding gene and *clpP-lntron1*, respectively. Notably, the majority of SSRs were found in *ycf1* gene in both *G. orientalis* and *G. officinalis* (Appendix A).

### 3.3. Adaptive Evaluation Analysis

The Ka/Ks ratio was used to evaluate the degree of selection constraint on each gene and estimate the selective pressure of protein-coding genes. Ka/Ks > 1 indicates the gene is under positive selection, Ka/Ks = 1 indicates the gene is under neutral selection and Ka/Ks < 1 indicates purification selection.

In the present study, KaKs Calculator was used to calculate the non-synonymous (Ka) to synonymous (Ks) ratio (Ka/Ks) of 71 effectively shared protein-coding genes in *G. orientalis* and species belonging to six other genera (Appendix A). The results showed that the Ka/Ks values between *G. orientalis* and Galegeae species ranged from 0 (*petN*) to 1.33757 (*petL*). Among them, the Ka/Ks value of *rpl20* gene between *G. orientalis* and *G. officinalis* was >1 (1.00749). However, the Ka/Ks values of *rpl20* among *G. orientalis* and other species of Galegeae were all lower than 0.5 (Appendix A). The Ka/Ks values of *petL* and *ycf4* genes between *G. orientalis* and *Calophaca* were >1 (1.33757, 1.02613). *Caragana ycf4* gene ratio was also >1 (1.04846). These results indicated that *rpl20* gene was positively selected within *Galega*, but was under strong purification selection pressure in other Galegeae genera. *petL* and *ycf4* genes were positively selected between *G. orientalis* and *Calophaca*. The *ycf4* gene evolved under beneficial mutations between *G. orientalis* and *Caragana* (Figure 5).

### 3.4. Sequence Divergence Analysis

The highly variable Cp genome is not only useful for identifying closely related species, but also can provide abundant phylogenetic information for evolutionary research. We compared the cp genomes of *G. orientalis* and 14 other Galegeae species. The results showed that the cp genome size in Galegeae species ranged from 122,210 bp (*Oxytropis diversifolia*) to 129,331 bp (*Caragana korshinskii*) and GC content ranged from 33.94% (*Astragalus galactites*) to 34.68% (*Tibetia liangshanensis*). *G. orientalis* differed significantly from other species in the length of the CDS and non-coding regions (Appendix A). We used mVISTA to compare the cp genomes of 14 species of Galegeae with *G. orientalis* as a reference. In general, *G. orientalis* and *G. officinalis* showed highly sequence similarity in the coding regions, and only *ycf1* and *rpl23* genes were found to be significantly different and highly variable. In contrast, the sequence variation was significantly higher in conserved non-coding sequence regions (CNS) than in other regions. In addition to *ndhD-ccsA*, *ndhE-psaC*, *petB-petD*, *trnS-GGA-ycf3*, *trnK-UUU-matK* and *psaJ-rpl33*, almost all CNS regions showed variation. In addition, we found a high degree of sequence variation in the *accD*, *rpoC2* and *ycf4* genes. In the UTR region of the *rrn23* gene, the degree of regional variation in the CNS was similar between *G. orientalis* and *G. officinalis* but significantly different among 14 species. Overall, the degree of sequence variation in conserved non-coding sequence regions was significantly higher than that in the coding regions (Figure 6).

We also calculated *p*-values between *G. orientalis* and *G. officinalis* to further clarify the extent of nucleotide variation within these species (Appendix A). In general, the *p*-values of all genes in *Galega* were low, and only the *clpP* gene exceeded 1, indicating high nucleotide variation. The *clpP* gene could provide a wealth of information about the phylogeny of Galegeae species.

### 3.5. Phylogenetic Relationships Based on Chloroplast Genomes

To identify the phylogenetic position of *G. orientalis* in the Papilionoideae subfamily, we used 31 complete chloroplast genomes of Papilionoideae subfamily species (15 Galegeae species) for phylogenetic analysis (Figure 7A), with *Pterocarpus santalinus* and *Arachis hypogaea* as outgroups. The results showed the seven genera of Galegeae were differentiated, with bootstrap support values of most nodes >95. *G. orientalis* and *G. officinalis* were clustered into one clade, and were more closely related to *Lotus corniculatus*.

In addition, we selected two genes (*ycf1* and *clpP*) with strong sequence differences and high Pi values to construct phylogenetic trees of 15 Calegeae species by comparing chloroplast genomes of *G. orientalis* and *G. officinalis*. The results showed that the phylogenetic topology of Galegeae species based on *ycf1* (Figure 7B) and *clpP* (Figure 7C) was almost the same as the phylogenetic tree constructed using the complete chloroplast genome. Seven Galegeae genera were clearly discriminated based on *ycf1* and *clpP* genes. *Tibetia liangshanensis* and *Galega* species (*G. orientalis* and *G. officinalis*) were grouped together in one cluster (Figure 7B). However, as shown in Figure 7C, *T. liangshanensis* was clustered in a separate clade. Furthermore, all three evolutionary trees placed *Calophaca sinica* in the same group as *Caragana* (*C. jubata* and *C. korshinskii*), suggesting that they are more closely related.

## 4. Discussion

### 4.1. General Features of G. orientaliscp Genome

In the present study, we sequenced and assembled the complete chloroplast genome of *G. orientalis*, which was in a length of 125,280 bp and structurally similar to other Papilionoideae species without an IR region [40]. The chloroplast genomes of *G. orientalis* and *G. officinalis* are highly conserved and similar in gene composition, structure and GC content, showing close species relationships within *Galega*. Regarding genetic differences, one intron in the *ycf1* gene was found deleted in the chloroplast genome of *G. officinalis* but present in *G. orientalis*. The *ycf1* gene length of *G. orientalis* is 4128 bp with one 30-bp intron and two exons: exon1 and exon2 with length of 1881 and 2217 bp, respectively. There is no intron in *ycf1* gene in other known Galegeae chloroplast genomes. The reason for the existence of an intron in *ycf1* gene in Papilionoideae cp genome has not been reported. In land plants, introns are crucial for promoting gene transcription [41]. We found a short intron in *ycf1* gene of *G. orientalis*, indicating that the structural variation of intron in *ycf1* gene of *G. orientalis* can be used as a molecular marker to provide important information at low taxonomic level in phylogenetic studies of leguminous species [42]. The *clpP* gene also showed loss of introns. Studies have shown that *clpP* genes of most species in the IRLC lineage of Papilionoideae have lost one intron, and only *Glycyrrhiza glabra* has lost two introns. In the present study, the loss of one intron in the *clpP* gene of *G. orientalis* and *G. officinalis*, and two introns lost in the *clpP* gene of *Glycyrrhiza zaissanica*, *Glycyrrhiza gobica* and *Glycyrrhiza alaschanica*, indicating that introns lost is a common phenomenon could occur in *Glycyrrhiza* species [43,44]. In addition, we also found the loss of two introns in the *clpP* gene of *Tibetia liangshanensis* (Appendix A).

Furthermore, we identified 74 protein-coding genes in the chloroplast genome of *Galega orientalis*. Compared with *G. officinalis, G. orientalis* cp genome was found lacked five unique genes, namely *trnE-UUC*, *rps16*, *rpl22*, *ycf2* and *pbf1*. This may be due to the nuclear translocation of these genes during the evolution of *G. orientalis*. A similar result was also revealed in the cp genome of *Cicer arietinum* with deletion of the *rpl22* gene [44]. Among them, *pbf1* gene is also absent in cp genome of other Galegeae species, but present in cp genome of *G. officinalis*. The reason may be that the deletion of IR region has leaded to gene rearrangement. Numerous studies have shown that IRLC lineage often results in gene deletion due to gene rearrangement, such as loss of *accD/rpl2* genes in some species of *Trifolium* [14], loss of *rpl23* gene in *Lathyrus* and *Vicia* [45], absence of *atpF* gene in *Cistanche* [46] and loss of *psbN* gene in *Haloxylon* [47]. The lack of IR region, size variation of IR region and many tandemly repeated sequences could cause chloroplast rearrangements, genetic structural variation or deletions.

Compared with angiosperms with intact chloroplast tetrad structure, *Galega* has fewer protein-coding genes, which may be caused by the deletion of IR region. Five of these genes (*accD*, *ycf1*, *ycf2*, *rpl23*, *infA*) were found only in certain species like *Trifolium* and *Medicago* [48]. *InfA*, which is considered to be the most transferable gene in the cp genome, and only exists in 24 angiosperms lineages, was absent in the cp genome of any known Galegeae species [49].

### 4.2. SSRs and Tandem Repeat Analysis

In the present study, we found high similarity in the types and number of tandem repeats between *G. orientalis* and *G. officinalis.* In addition, forward repeat was the most common type and most repeats were distributed in the IGS region, suggesting their potential as genetic markers for phylogenetic studies [50]. As the chloroplast genome is characterized by maternal inheritance and possess low frequency of genetic recombination, cp simple repeats (SSRs) can be used as effective molecular markers for genetic diversity research, species identification and phylogenetic analysis [51]. We observed abundant mononucleotide repeats in *G. orientalis* and *G. officinalis* chloroplast genomes with A/T motif repeats being dominant. An extremely strong A/T bias in SSR loci has also been observed in other legumes such as *Vigna radiate* [52] and *Arachis hypogaea* [53], which may result in base composition bias. At present, the potential role of cpSSRs in ecology and plant evolution has not been explored. Therefore, the cpSSRs detected in the two species could be used to assess the genetic relationships between different species and detect the polymorphisms in the Galegeae species at the population level.

### 4.3. Sequence Variation Analysis and Hotspots

Chloroplast genome is useful for the study of species evolution and classification. The CNS showed more obvious hypervariable regions than the other regions when *G*. *orientalis* was compared with 14 other Galegeae species. In the protein-coding region (CDS), only *ycf1* and *rpl23* genes showed strong differences. At the same time, *ycf1* gene had a higher Ka/Ks value, suggesting that it is under less purifying selection pressure. The accelerated evolution rate of *petL* gene was observed in *Galega*. The *ycf4* gene has undergone adaptive evolution between *G. orientalis* and *Calophaca*/*Caragana*. It is also found in *Lathyrus*, *Pisum* and *Vavilovia* [54,55]. However, high sequence variation was found in the CNS regions except *ndhD-ccsA*, *ndhE-psaC*, *petB-petD*, *trnS-GGA-ycf3*, *trnK-UUU-matK* and *psaJ-rpl33*. These hypervariable genomic regions were considered to be useful markers for elucidating the phylogenetic relationships among the Galegeae species. The *ycf1* gene also showed potential as a genetic marker for *Quercus bawanglingensis* [56]. *atpB-rbcL* was used to assess the germplasm differentiation of *P. lunatus* [57], *trnW-CCA-petG* and *psbN-psbH* were also used for phylogenetic and domestication studies of *Astragalus* species [58]. The polymorphism of chloroplast genome is useful for evolutionary analysis of *G. orientalis*. Furthermore, we found that the *clpP* gene had the highest Pi value in the chloroplast genomes of *G. orientalis* and *G. officinalis* and could be used for phylogenetic analysis and population genetic study of *Galega* species.

### 4.4. Phylogenetic Relationships

Based on complete chloroplast genome sequences of 31 legume species, we determined the position of the genus in the Papilionoideae using phylogenetic analysis, and the topological structure was relatively consistent. *G. orientalis* and *G. officinalis* are closely related, clustered into one group, and were grouped in the same clade with *Lotus corniculatus*. These results indicate that besides the *Galega* family species, the *Lotus* also has a close evolutionary relationship with the *G. orientalis*. However, in Galegeae, the species of the *Tibetia* did not belong to the same clade as other genera, which was also confirmed in the phylogenetic tree constructed based on *clpP* gene. This may be due to the absence of IR region in the ancestral species of the Galegeae species, resulting in a series of alternations (including high rearrangements and duplications) during evolution, leading to differences among congeners [58].

## 5. Conclusions

In this study, the complete chloroplast genome of *Galega orientalis* was sequenced and annotated, and compared with the cp genome of other Galegeae species. The results showed that the length and structure of the chloroplast genome of *G. orientalis* were similar to those of *G. officinalis*, but the length of the non-coding region varied significantly. mVISTA analysis results showed that the mutation rate of the non-coding region was higher in *G. orientalis* than other Galegeae species, and the coding region was more conserved. Some hypervariable regions were identified, which could be used to distinguish species. In addition, we identified two hotspot genes (*ycf1* and *clpP*) that contained rich information for species identification and phylogenetic reconstruction of Galegeae, as well as provide insights into the evolution of Galegeae species.

## Figures and Tables

**Figure 1 genes-14-00176-f001:**
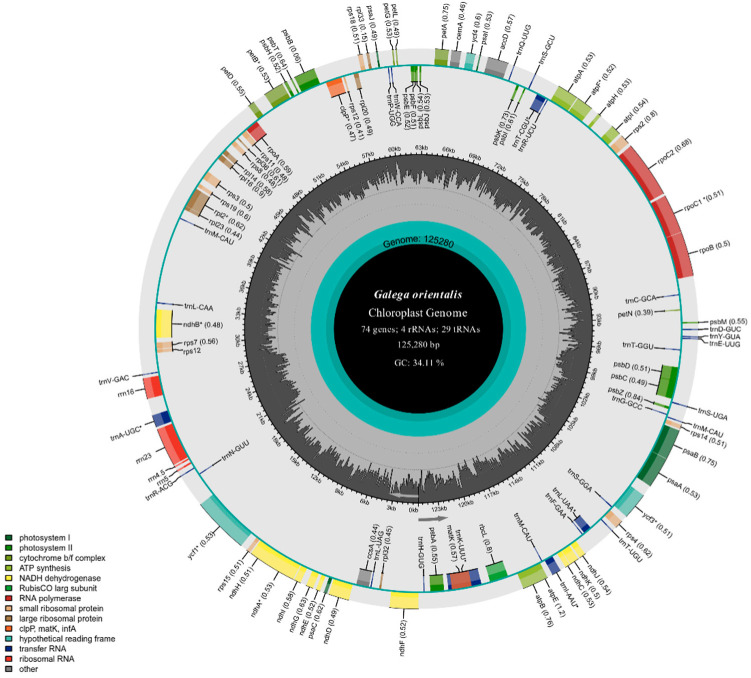
The chloroplast genome map of *Galega orientalis*. The genes drawn inside and outside of the circle are transcribed counterclockwise and clockwise, respectively. The dark gray inside the circle represents GC content, and the light gray corresponds to AT content. * Genes with introns.

**Figure 2 genes-14-00176-f002:**
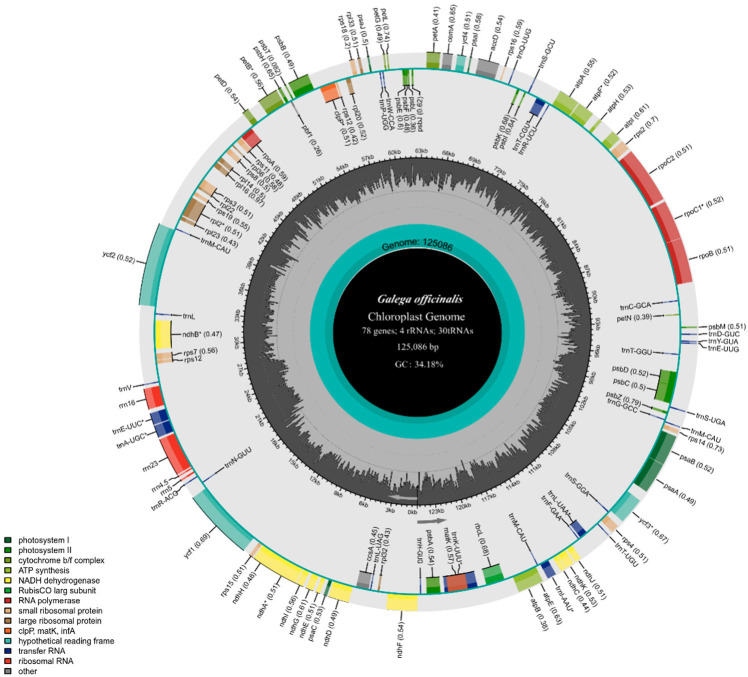
The chloroplast genome map of *Galega officinalis*. The genes drawn inside and outside of the circle are transcribed counterclockwise and clockwise, respectively. The dark gray inside the circle represents GC content, and the light gray corresponds to AT content. * Genes with introns.

**Figure 3 genes-14-00176-f003:**
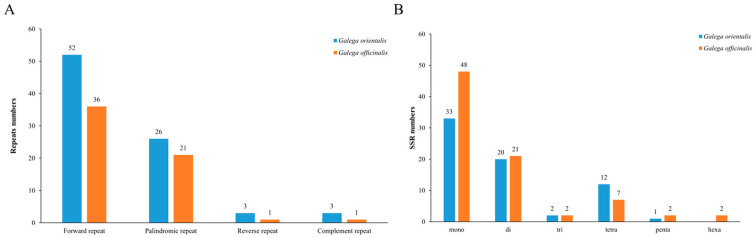
Analysis of repeat sequences in the chloroplast genome of *G. orientalis* and *G. officinalis* (**A**). SSR analysis of chloroplast genome of *G. orientalis* and *G. officinalis* (**B**).

**Figure 4 genes-14-00176-f004:**
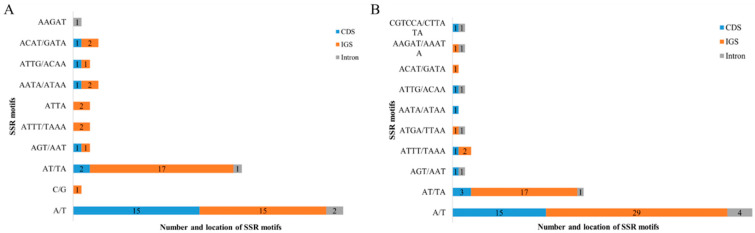
The types and distribution of SSRs along the chloroplast genomes of *G. orientalis* (**A**) and *G. officinalis* (**B**). Different locations, including CDS, IGS and intron regions, are represented as colored boxes.

**Figure 5 genes-14-00176-f005:**
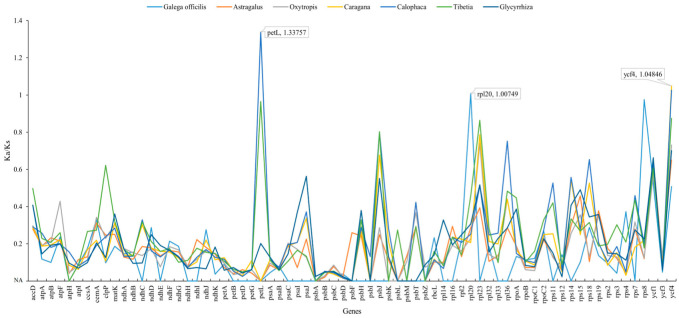
The Ka/Ks values of 71 shared genes between *G. orientalis* and other Galegeae species. The Ka/Ks values between *G. orientalis* and each genus are the average values of the Ka/Ks between *G. orientalis* and the species within the genus.

**Figure 6 genes-14-00176-f006:**
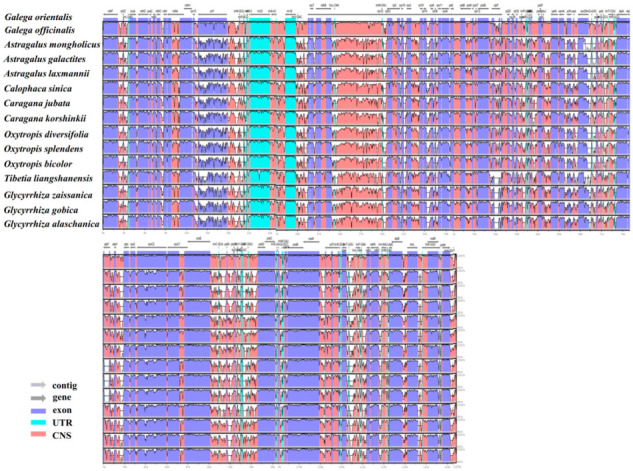
The comparison of 15 Galegeae species cp genomes by using mVISTA. *G. orientalis* as a reference, genomic regions are color-coded to indicate protein-coding regions, exons, UTRs, and CNS.

**Figure 7 genes-14-00176-f007:**
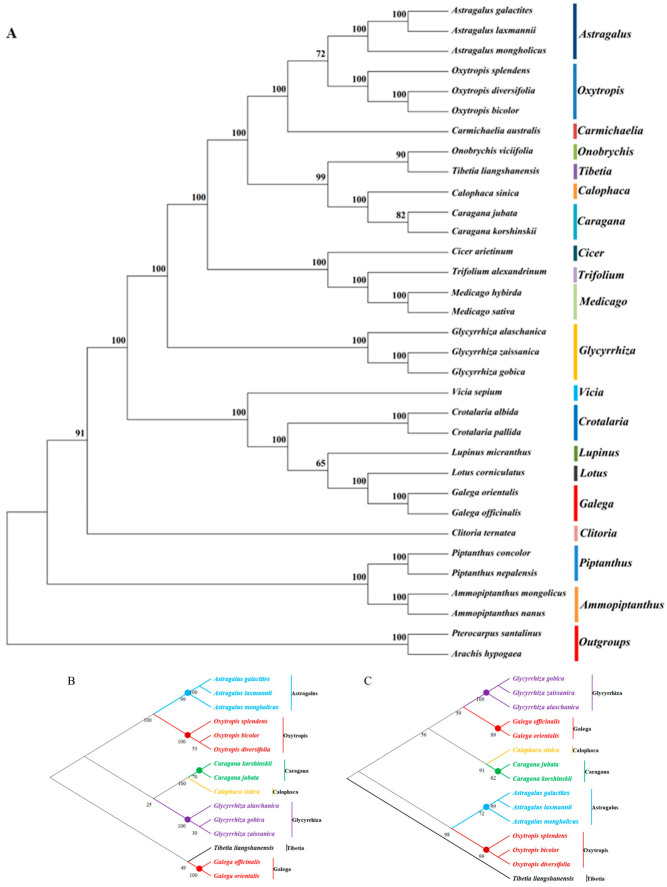
(**A**) Phylogenetic tree of the Leguminosae species (*Galegeae*) based on 31 complete chloroplast genomes and constructed using the *ycf1* gene (**B**) and *clpP* gene (**C**), with the *Galega* species indicated in red.

**Table 1 genes-14-00176-t001:** Lengths of introns and exons of the split genes in the *Galega orientalis* complete chloroplast genome.

Gene Name	Gene Location			Length (bp)				
	Strand	Start	End	Exon I	Intro I	Exon II	Intro II	Exon III
*ndhA*	−	9250	11,603	553	1262	539		
*ycf1*	−	13,671	17,798	1881	30	2217		
*trnA-UGC*	−	24,166	25,050	37	812	36		
*ndhB*	+	30,629	32,792	721	685	758		
*rpl2*	+	41,007	42,519	391	688	434		
*petB*	−	51,034	52,517	9	827	648		
*clpP*	+	55,465	56,683	360	631	228		
*trnT-CGU*	+	71,994	72,731	35	660	43		
*atpF*	−	74,721	75,956	145	681	410		
*rpoC1*	−	84,108	86,933	430	771	1625		
*ycf3*	−	106,655	108,601	124	719	232	721	151
*trnL-UAA*	+	111,044	111,688	35	540	50		
*trnI-AAU*	−	114,670	115,325	32	566	58		
*trnK-UUU*	+	120,448	122,994	37	2475	35		

**Table 2 genes-14-00176-t002:** Genes predicted in the chloroplast genome of *Galega orientalis*.

Category	Function	Name of Genes					
Ribosomal proteins (11)	Small subunit of ribosome (SSU)	*rps15*	*rps7*	*rps19*	*rps3*	*rps8*	*rps18*
		*rps2*	*rps14*	*rps4*	*rps11*	*rps12*	
Transcription (12)	Large subunit of ribosome (LSU)	*rpl32*	*rpl23*	*rpl2*	*rpl16*	*rpl14*	*rpl36*
		*rpl20*	*rpl33*				
	RNA polymerase subunits	*rpoA*	*rpoC1*	*rpoC2*	*rpoB*		
Self-replication (33)	Ribosomal RNA Genes	*rrn5*	*rrn4.5*	*rrn23*	*rrn16*		
	Transfer RNA genes	*trnL-UAG*	*trnN-GUU*	*trnR-ACG*	*trnA-UGC*	*trnV-GAC*	*trnL-CAA*
		*trnM-CAU* ^a^	*trnP-UGG*	*trnW-CCA*	*trnQ-UUG*	*trnS-GCU*	*trnT-CGU*
		*trnR-UCU*	*trnC-GCA*	*trnD-GUC*	*trnY-GUA*	*trnE-UUC*	*trnT-GGU*
		*trnS-UGA*	*trnG-GCC*	*trnM-CAU*	*trnS-GGA*	*trnT-UGU*	*trnL-UAA*
		*trnF-GAA*	*trnI-AAU*	*trnM-CAU*	*trnK-UUU*	*trnH-GUG*	
Photosynthesis related genes (44)	Subunits of Photosystem I	*psaJ*	*psaI*	*psaB*	*psaA*	*psaC*	
	Subunits of Photosystem II	*psbH*	*psbT*	*psbB*	*psbK*	*psbI*	*psbC*
		*psbE*	*psbF*	*psbL*	*psbJ*	*psbM*	*psbZ*
		*psbA*	*psbD*				
	Large subunit of Rubisco	*rbcL*					
	Subunits of ATP synthase	*atpA*	*atpF*	*atpH*	*atpI*	*atpE*	*atpB*
	Cytochrome b/f complex	*petD*	*petB*	*petG*	*petL*	*petA*	*petN*
	C-type cytochrome synthesis gene	*ccsA*					
	Subunits of NADH dehydrogenase	*ndhF*	*ndhD*	*ndhE*	*ndhG*	*ndhI*	*ndhA*
		*ndhH*	*ndhB*	*ndhJ*	*ndhK*	*ndhC*	
Other genes (7)	Maturase	*matK*					
	Protease	*clpP*					
	Chloroplast envelope membrane protein	*cemA*					
	Subunit of acetyl-CoA	*accD*					
	Hypothetical open reading frames	*ycf1*	*ycf3*	*ycf4*			

Note: ^a^ duplicated gene.

## Data Availability

The data presented in this study are available on request from the corresponding author. The data are not publicly available due to privacy. The complete chloroplast genome sequence of *G. orientalis* was deposited at CNGBdb (accession number CNP0003740).

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
