# Peer review of "Analysis of Complete Chloroplast Genome: Structure, Phylogenetic Relationships of Galega orientalis and Evolutionary Inference of Galegeae"

_genes, 2023, doi:10.3390/genes14010176_

Round 1
Reviewer 1 Report
Authors have sequenced the chloroplast genome of G. orientalis, which is a leguminous plant used for different purposes. They have compared it with other closely related species, G. officinalis (which is already sequenced). They identified some genes which could be used as markers for the study of evolutionary analysis of these species.
They have written the manuscript well and have provided all the necessary information for the validation of their study. There are few minor corrections and grammatical errors that need to be fixed. Detailed suggestions and comments are provided in the word document.

Reviewer 2 Report
Review of Analysis of complete chloroplast genome: structure, phylogenetic relationship of Galega orientalis and evolutionary inference of Galegeae by Feng et al. for Genes
[I've also included this as a pdf in case this renders weird]
I am recommending this manuscript for “Resubmission after major revision.” There are omissions or possible issues with the phylogenetic analysis and with how the phylogenetic analysis results are displayed. There are some other issues with the Ka/Ks analysis. Furthermore, this manuscript was difficult to read and I suggest further proofreading and a thorough check for grammatical errors.
Notably there was much confusion over singular and plural words in this paper; there was a noticeable error in the title: “relationship” should be “relationships”
Here are some issues with the substance of this paper:
How was GTR+Gamma chosen as a molecular evolution model? Was there some model fitting like that done in jmodeltest performed?
Arabidopsis seems like too distant of a choice for outgroup.
It was good that Ka/Ks values were calculated but some values, like that for petN, were based on very few substitutions. The values for petN, for example, having zero nonsynonymous substitutions may be sampling error rather than a particularly strong indication of purifying selection. It would be good to note this or include uncertainty or error estimates of some kind on some of these values.
Figure 7 includes only a cladogram. A phylogram (a phylogenetic tree with meaningful branch lengths) should be included as part of a main figure – including just a cladogram, like this in this paper, can mask many of the potential issues with the analysis (like potential long branch attraction, inappropriate outgroup, etc.).
There may have been other issues but, due to issues with this manuscript’s grammar and readability, the details of the analysis were unclear at times.

Round 2
Reviewer 2 Report
Review of Analysis of complete chloroplast genome: structure, phylogenetic relationship of Galega orientalis and evolutionary inference of Galegeae by Feng et al. for Genes (2nd round of review)
[I've also included this as a pdf in case this renders weird]
Many of the changes made to address my comments exceeded expectations but the most important substantive issue (lack of a phylogram in the main text) is not satisfactorily addressed. This is a superficial criticism in isolation, so I will recommend this manuscript for acceptance after minor revisions, but a phylogram may reveal other issues; so it is an essential change. The changes will be addressed in dialogue with the authors’ responses to my criticisms below:
Response 1: Thanks for the reviewer’s comments. We have carefully checked the singular and plural words in the paper and corrected them.
There are still some issues with this but they seem so few that hopefully they will be addressed in the copyediting phase.
Response 2: Thanks for the reviewer’s comments. I'm sorry that because we only checked the literatures to determine the model and did not test whether it is suitable for our research. We have given up using the model of “GTR+Gamma” now. We have re-identified the species involved in phylogenetic analysis and detected the optimal molecular evolution model to be TVM+F+R7 by IQ- TREE software's ModelFinder module. The following are the references for my use of this method.
Excellent—this is complete and sufficiently descriptive.
Response 3: Thanks for the reviewer’s comments. According to the reviewer’s comments, we have chosen Pterocarpus santalinus and Arachis hypogaea as new outgroups, and built a new phylogenetic tree.
Great choices – the use of two taxa for the outgroup is good here.
Response 4: Thanks for the reviewer’s comments. I'm sorry that we got the wrong results due to my wrong description. In our study, we have checked Ka and Ks values of each species and confirmed that the non synonymous substitutions were not zero but NA. In the calculation of KA/KS values, we incorrectly described it as 0. Now we have changed and deleted the results of petN gene. In addition, we have added the error estimate of KA/KS values in Table S4.
Great – this works and good call with petN. Going through table S4, I see the standard deviations and I see specific significances, which definitely answer these questions of sample size and uncertainty together.
Some Ka/Ks values are far from significant, are they Bonferroni corrected? If so, this should be stated in the text.
Response 5: Thanks for the reviewer’s comments. We have selected new species to participate in the phylogenetic tree construction and selected Pterocarpus santalinus and Arachis hypogaea as new outgroups, which may reduce some potential problems in the analysis.
I think that the ways that you revised the analysis likely had a positive impact but without a phylogram, it is difficult to ascertain or survey some potential issues. This is the most important part of this critique; a phylogram needs to be included in the main text of the paper.
Response 6: Thanks for the reviewer’s comments. According to the reviewer’s comments, we have corrected the grammar and spelling problems in the paper, which we believe will be more favorable for your review.
The paper is much improved and hopefully final copy-editing will improve any small outstanding issues.
